# Cell-based and isoform-selective G protein-coupled receptor kinase assays for comprehensive inhibitor evaluation
Nina K. Blum[1], Manuela C. Kiefer[1], Angelika Decker[1], Laura Klement [2], Edda S. F. Matthees [2], Verena Weitzel[2], Falko Nagel[3], Babu Joseph[4], Julia Drube [2], David Uehling[4], Carsten Hoffmann [2] & Stefan Schulz [1,3] ✉

G protein-coupled receptor (GPCR) signaling is regulated by four ubiquitously expressed GPCR kinase isoforms (GRKs), namely GRK2, GRK3, GRK5, and GRK6. Overexpression of individual GRKs occurs in diseases like cancer and heart failure, prompting a search for potent GRK inhibitors. While various in silico and in vitro approaches exist, few methods assess inhibitor efficacy in cellular systems. To address this, we used HEK293 cell lines co-expressing the β2 adrenergic receptor (β2) and one GRK isoform on a quadruple GRK2/3/5/6 knockout background (ΔQ-GRK). We evaluated the inhibition of isoproterenol (ISO)-induced T360/S364-β2 phosphorylation using the 7TM phosphorylation assay. This combination allowed comprehensive evaluation of commercially available GRK inhibitors. We conclude that compound 8h (GRK2/3 inhibitor) and compound 18 (GRK5/6 inhibitor) are highly recommendable tools for the study of GPCR phosphorylation and function in cellular systems. Together, these cell-based GRK inhibitor assays can facilitate medium- to high-throughput screening of future GRK-targeted drug candidates.

G protein-coupled receptors (GPCRs) constitute a large family of proteins with more than 800 members distributed throughout the human body[1]. Their involvement in various physiological and pathophysiological processes makes them pharmacological targets for therapeutic intervention. These membrane proteins act as cellular sensors by responding to a variety of extracellular stimuli and mediating intracellular signaling pathways through the activation of G proteins. This generally results in the transmission and amplification of signals throughout the cell. To prevent GPCR hyperactivity and cell overstimulation, desensitization and downregulation of these receptors must be tightly controlled. One aspect is the ligand-induced receptor phosphorylation by GPCR kinases (GRKs), which subsequently facilitates β-arrestin binding. This adapter protein association blocks further receptor-G protein interactions and initiates internalization processes instead. Thus, GRK-catalyzed receptor phosphorylation and β-arrestin binding terminate GPCR activity, which contributes to the maintenance of cell homeostasis[2].

The GRK family consists of only seven serine/threonine protein kinases, which stands in contrast to the vast number of GPCRs they regulate. Among them, GRK1, GRK4, and GRK7 show a restricted distribution in tissues such as retina or testis, whereas GRK2, GRK3, GRK5, and GRK6 are

expressed more ubiquitously[3,4]. Both GRK2 and GRK3, as well as GRK5 and GRK6, exhibit overlapping substrate specificities within their respective subfamilies. But the number and location of potential GRK target phosphorylation sites vary considerably among GPCRs, and information obtained from studies using putative "model" receptors cannot be easily extrapolated to other receptors. Furthermore, although ligand-bound GPCRs are their primary substrate, GRKs can also exert non-canonical functions by phosphorylating cytosolic proteins and are able to modulate other signaling pathways independently of their kinase activity[5–7]. A dysregulation in GRK expression has been implicated in a variety of disorders ranging from cancer, cardiovascular and metabolic pathologies, to inflammation, neurodegeneration as well as opioid tolerance and addiction[8–19].

Therefore, GRK inhibitors have received much attention in the last decade. First, they can be used as tool compounds to dissect the role of GRKs during receptor phosphorylation, thus contributing to a better understanding of the signaling pathways of clinically relevant GPCRs. Second, pharmacological inhibition of GRKs improved disease outcomes in animal studies, suggesting them as potential future treatment options[20–27]. Ongoing research has led to the development of novel small-molecule inhibitors with increased selectivity and potency. Compound 101 has already been used

[1]Institut für Pharmakologie und Toxikologie, Universitätsklinikum Jena, Friedrich-Schiller-Universität Jena, Jena, Germany. [2]Institut für Molekulare Zellbiologie, CMB—Center for Molecular Biomedicine, Universitätsklinikum Jena, Jena, Germany. [3]7TM Antibodies GmbH, Jena, Germany. [4]Drug Discovery Program, Ontario Institute for Cancer Research, Toronto, ON, Canada. ✉e-mail: stefan.schulz@med.uni-jena.de

extensively to study the influence of GRK2/3[28–33]. Examples of recently developed GRK2 inhibitors include the paroxetine analog CCG258747 and compound 8h, both of which have also been tested in vivo[34,35]. CCG273441, originally based on the structure of a receptor tyrosine kinase inhibitor, is unique in that it binds covalently to GRK5[36]. In addition, compounds 18 and 19 have recently been described as selective GRK6 inhibitors[37].

To characterize these inhibitors, previous studies have used several well-established in silico and in vitro approaches. Although these methods provide critical information on the mode of action and binding affinity of these compounds, few provide evidence of their efficacy in cellular systems. To this end, we used genetically engineered HEK293 cell lines co-expressing the β2 adrenergic receptor (β2) and one GRK isoform on a quadruple GRK2/3/5/6 knockout background (ΔQ-GRK). We then evaluated and compared the effects of several commercially available GRK inhibitors on isoproterenol (ISO)-induced β2 phosphorylation at the T360/S364 site. Therefore, this assay allowed assessment of GRK inhibitor efficacy in living cells, complementing previous in vitro and in silico approaches. We further selected some of these inhibitors to demonstrate their utility in other receptor systems and functional readouts. Our work provides a comprehensive assessment of currently available GRK inhibitors and presents adapted cell-based immunoassays to screen for future potential candidates.

## Results

### Screening of commercially available GRK inhibitors using GRK knockout cell lines

The aim of this work was to establish a toolkit for rapid *in cellulo* screening of GRK inhibitors. For this purpose, we chose the β2, since this receptor is regulated by various non-visual GRK isoforms[38,39]. Phosphorylation of this receptor was assessed using a phosphosite-specific antibody directed against the T360/S364 residues in the 7TM phosphorylation assay established by Kaufmann, et al.[40]. In this study, we focused on one GRK-dependent phosphorylation site of the β2. Preliminary analyses of pS355/pS356 revealed similar agonist-dependent phosphorylation profiles, suggesting that analysis of this additional site would not have provided further relevant insights for the aims of this study. Furthermore, the antibody against pT360/S364 yielded a superior signal-to-noise ratio allowing robust assessment of receptor phosphorylation (Supplementary Fig. 2b). In initial experiments, we used HEK293 cells expressing the GRK isoforms 2, 3, 5, and 6 at endogenous levels (Control), double-knockout cell lines of GRK isoforms 2

and 3 (ΔGRK2/3) or 5 and 6 (ΔGRK5/6), as well as ΔQ-GRK cells lacking expression of all four ubiquitously expressed GRKs[41] that stably express the β2 (Fig. 1a). As shown in Figs. 1b and 1c, both double-knockout cell lines were able to generate a robust phosphorylation signal with increasing ISO concentrations, reaching ~65-70% of the signal in Control cells (Supplementary Table 1). Phosphorylation of T360/S364 could not be detected in ΔQ-GRK cells, indicating a solely GRK-dependent phosphorylation at these residues. On this basis, we used the ΔGRK2/3 and ΔGRK5/6 cell lines to characterize commercially available GRK inhibitors (Figs. 2 and 3). The pan-inhibitor LDC9728 served as a control for successful GRK inhibition[40]. First, we included inhibitors of GRK2 and GRK3, namely CCG258208, CCG258747, compound 8h, compound 101, and GSK180736A (Fig. 2, Supplementary Table 2). All compounds except GSK180736A completely blocked receptor phosphorylation in ΔGRK5/6 cells at a concentration of 30 μM. CCG258208, CCG258747, and compound 101 showed similar $IC_{50}$ values of log -5.0, while compound 8h was an order of magnitude more potent. In contrast, they were unable to inhibit GRK-mediated receptor phosphorylation in ΔGRK2/3 cells. The kinase inhibitor GSK180736A and its analog CCG215022 did not inhibit ISO-induced β2-phosphorylation in either cell line at the maximal concentration tested ($logIC_{50}$ above -4.5) (Supplementary Table 2). Second, we examined GRK5 and GRK6 inhibitors such as CCG273441, compound 10a, compound 18, compound 19, and compound 707 (Fig. 3, Supplementary Table 2). Out of these, CCG273441 blocked receptor phosphorylation in ΔGRK2/3 cells at a concentration of 100 nM, thereby exhibiting the lowest $IC_{50}$ (log -7.6) of the inhibitors tested. Additionally, it did not prevent receptor phosphorylation after agonist stimulation in ΔGRK5/6 cells. Compound 10a and its analogs compound 18 and compound 19 prevented the phosphorylation of β2 in ΔGRK2/3 cells with $IC_{50}$ values between log −5.4 and log −6.5. However, compound 10a also visibly blocked receptor phosphorylation of T360/S364 in ΔGRK5/6 cells at the maximal concentration tested ($logIC_{50}$ of −4.7), while its analogs did not. Compound 707 and KR-39038 were unable to induce a reduction of the phosphorylation signal in either cell line at any given concentration ($logIC_{50}$ above −4.5) (Supplementary Table 2).

### Differentiation of GRK specificities in cell systems over-expressing a single GRK isoform

With the ΔGRK2/3 and ΔGRK5/6 cells, we were already able to distinguish between two GRK subclasses. However, to further dissect the specificity of

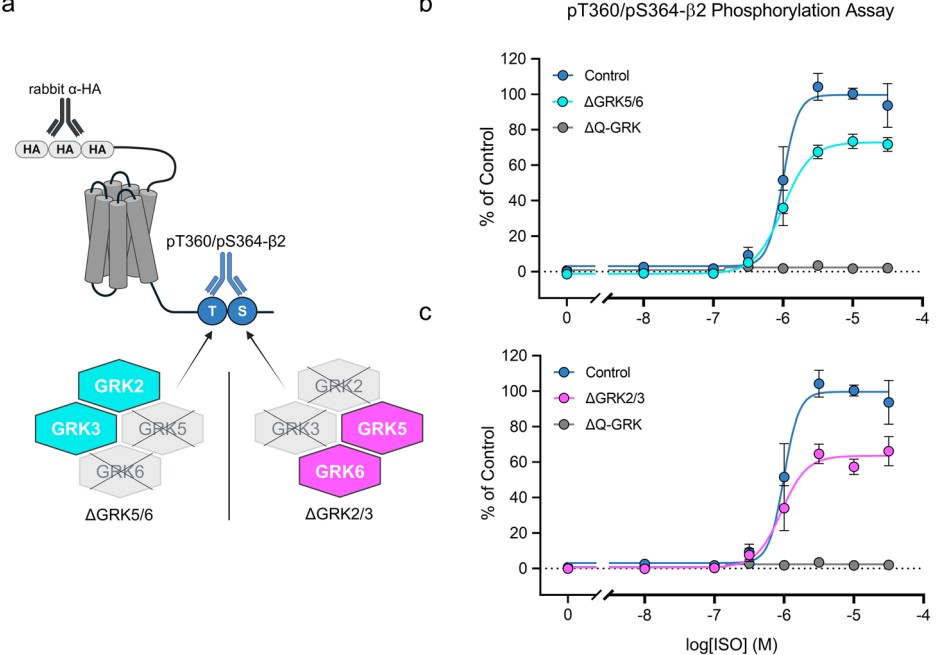

**Fig. 1 | Agonist-induced β2 adrenergic receptor (β2) phosphorylation in GRK knockout (ΔGRK) cell lines. a** Schematic representation of the β2 with antibody binding sites. The N-terminal hemagglutinin (HA)-tags allowed detection of the receptor independent of its phosphorylation status via anti-HA antibodies. Phosphorylation of the T360/S364 residues in the C-terminus was detected by a phosphosite-specific antibody. The β2 was expressed in cell lines exhibiting with a knockout of the GRK isoforms 5/6 (ΔGRK5/6) or 2/3 (ΔGRK2/3) (Created in BioRender. Blum, N. (2026) https://BioRender.com/d44ugtq). **b, c** HEK293 cell lines stably expressing the β2 were treated with increasing isoproterenol (ISO) concentrations (30 min, 37 °C) and phosphorylation of T360/S364 was assessed using the 7TM phosphorylation assay. Concentration-response curves were generated for ΔGRK5/6 (**b**) and ΔGRK2/3 (**c**). Cells expressing all endogenous GRKs (Control) or none of the GRKs (ΔGRK2/3/5/6, ΔQ-GRK) are shown as controls. All data were normalized to the optical density readings of Control. Graphs represent mean ± SEM of at least $n = 4$ independent experiments performed in duplicates.

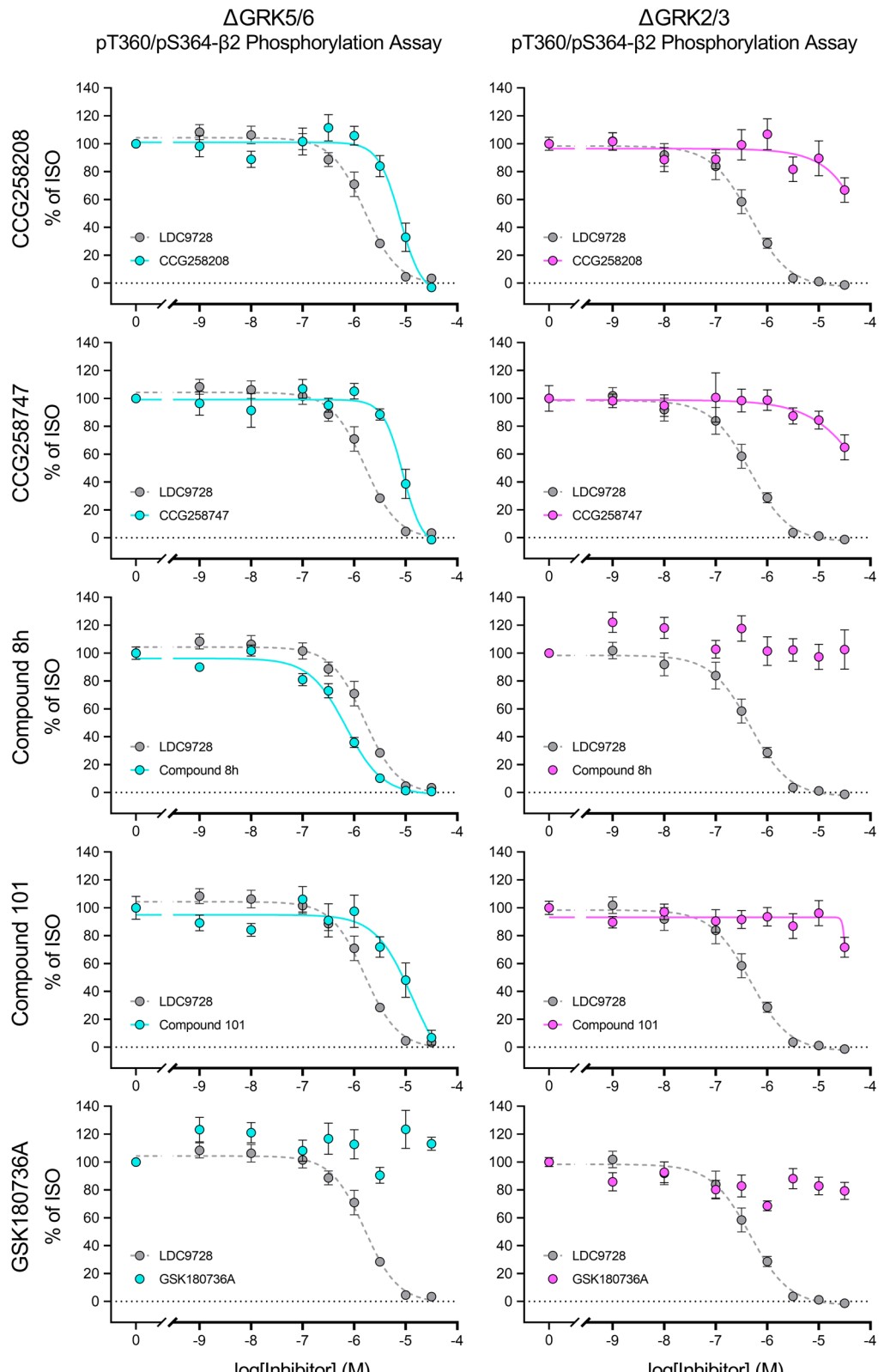

**Fig. 2 | Inhibition of GRK-mediated β2 adrenergic receptor (β2) phosphorylation by putative GRK2/3 inhibitors.** HEK293 ΔGRK5/6 (left) or ΔGRK2/3 (right) stably expressing the β2 were preincubated with increasing concentrations of the indicated GRK2/3 inhibitor (CCG258208, CCG258747, Compound 8h, Compound 101, GSK180736A) (30 min, 37°C) prior to treatment with 10 µM isoproterenol (ISO) (30 min, 37 °C). Receptor phosphorylation was assessed using the 7TM phosphorylation assay and data were normalized to 10 µM ISO without inhibitor. The pan-inhibitor LDC9728 (dashed in gray line) serves as reference in every graph. Data points represent mean ± SEM of $n$ = 5 independent experiments performed in duplicates.

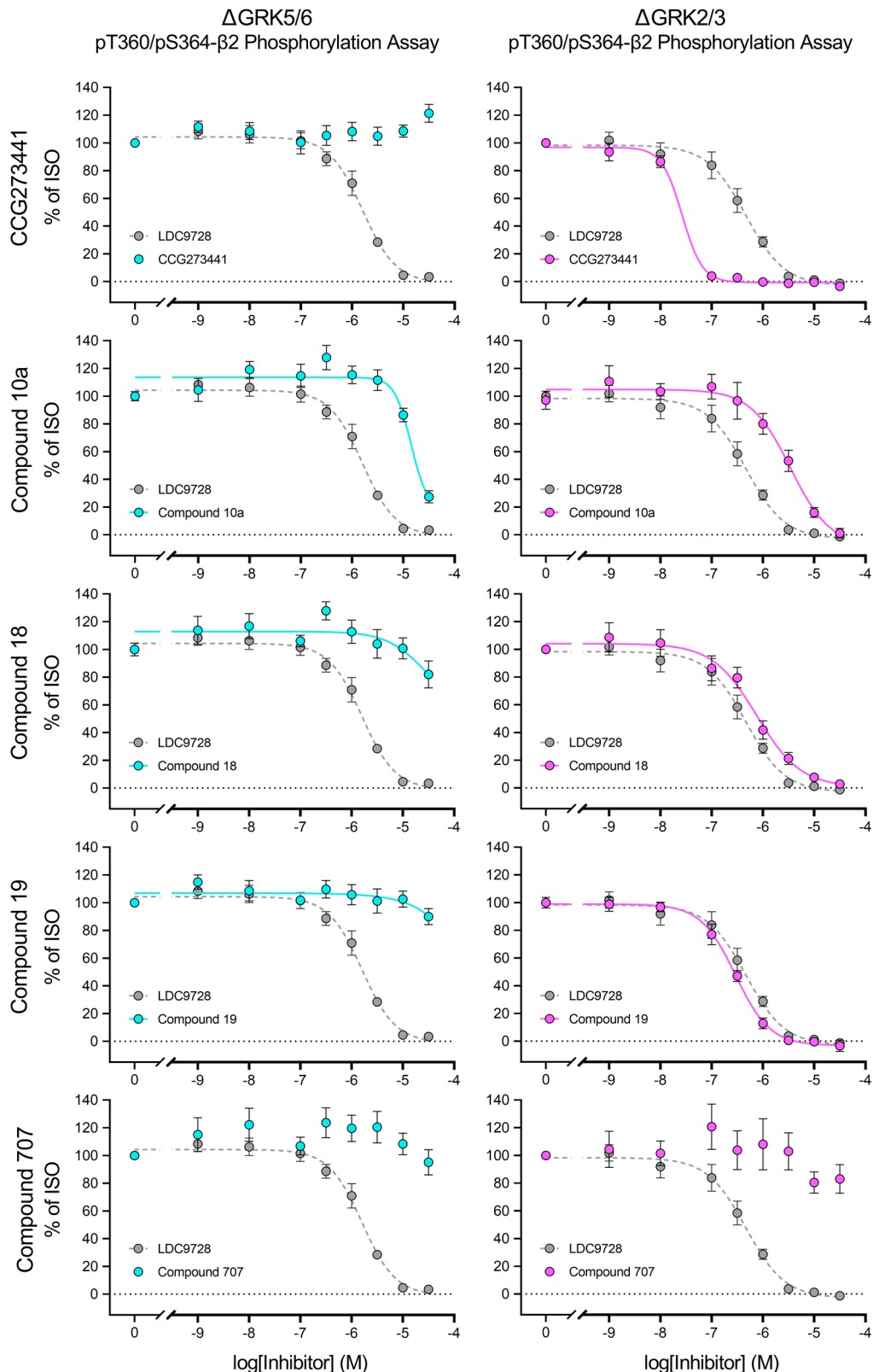

**Fig. 3 | Inhibition of GRK-mediated β2 adrenergic receptor (β2) phosphorylation by putative GRK5/6 inhibitors.** HEK293 ΔGRK5/6 (left) or ΔGRK2/3 (right) stably expressing the β2 were preincubated with increasing concentrations of the indicated GRK5/6 inhibitor (CCG273441, Compound 10a, Compound 18, Compound 19, Compound 707) (30 min, 37 °C) prior to treatment with 10 µM isoproterenol (ISO) (30 min, 37 °C). Receptor phosphorylation was assessed using the 7TM phosphorylation assay and data were normalized to 10 µM ISO without inhibitor. The pan-inhibitor LDC9728 (dashed in gray line) serves as reference in every graph. Data points represent mean ± SEM of $n = 5$ independent experiments performed in duplicates.

**Fig. 4 | Agonist-induced β2 adrenergic receptor (β2) phosphorylation in ΔQ-GRK cells over-expressing one GRK isoform. a** Schematic representation of the β2 with its antibody binding sites against the hemagglutinin (HA)-tag and an intracellular phosphorylation site. Cells express one of four GRK isoforms (ΔQ + GRK2 in red, ΔQ + GRK3 in green, ΔQ + GRK5 in yellow, ΔQ + GRK6 in purple) on a quadruplicate GRK2/3/5/6 knockout background (ΔQ-GRK in gray) (Created in BioRender. Blum, N. (2026) https://BioRender.com/dfnkeg7). **b** HEK293 cell lines stably expressing the β2 were stimulated with 10 μM isoproterenol (ISO) (30 min, 37 °C) (+) or were left untreated (−). Phosphorylation of the T360/S364 residues was imaged by Western blot analysis. Total receptor levels were checked with anti-HA antibody. (**c-f**) Cells were treated with increasing ISO concentrations (30 min, 37 °C). Receptor phosphorylation at T360/S364 was assessed with the 7TM phosphorylation assay and data was normalized to the optical density readings of Control. Graphs depict concentration-response curves in ΔQ + GRK2 (**c**), ΔQ + GRK3 (**d**), ΔQ + GRK5 (**e**) and ΔQ + GRK6 (**f**), each in comparison to signals produced in Control and ΔQ cells. Western blot image illustrates one representative of $n = 5$ replicates. Graphs display the mean ± SEM of at least $n = 5$ independent experiments performed in duplicates.

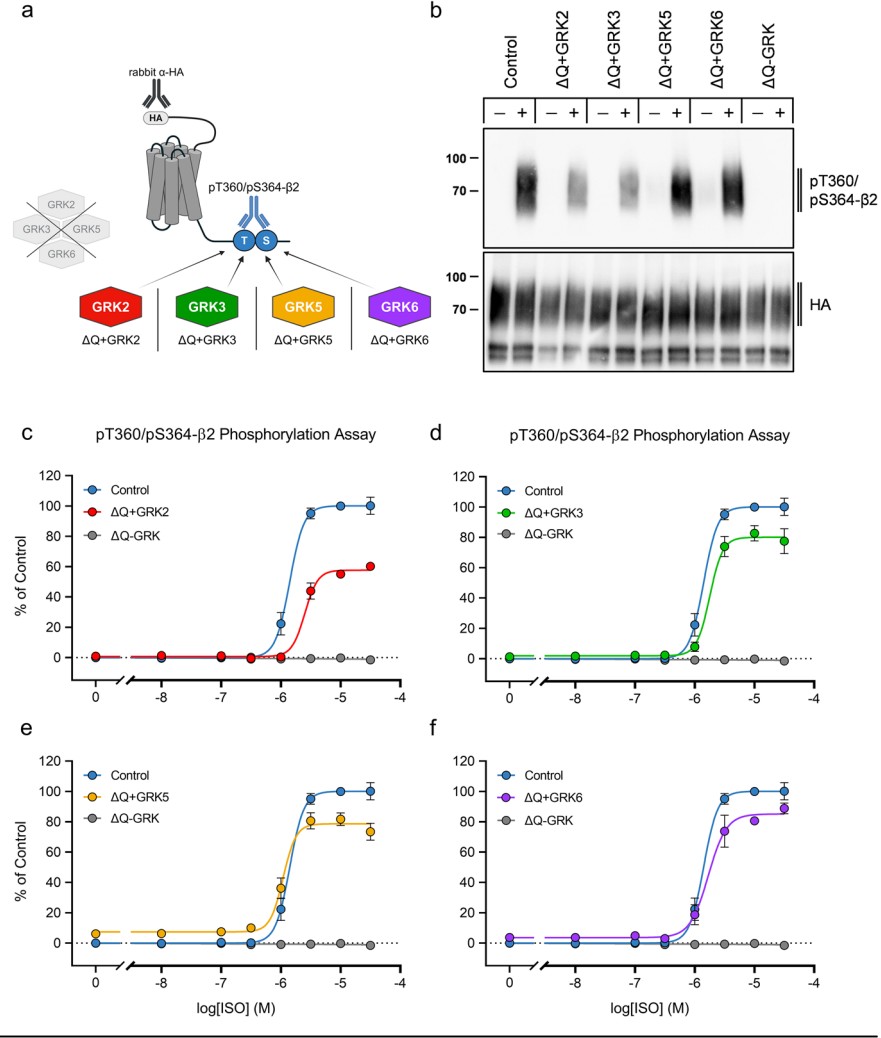

GRK inhibitors, we used ΔQ-GRK cells stably overexpressing only one of the GRK isoforms 2, 3, 5, or 6 (ΔQ + GRK2, ΔQ + GRK3, ΔQ + GRK5, and ΔQ + GRK6, respectively) (Fig. 4a). To check for receptor phosphorylation and total expression levels, we performed Western blot analysis (Fig. 4b). All GRK-expressing cell lines showed robust phosphorylation of T360/S364 upon ISO treatment (Fig. 4c–f). In concentration-response curves, ΔQ + GRK6 yielded the strongest receptor phosphorylation, reaching ~90% of the signal in Control cells, followed by ΔQ + GRK3 and ΔQ + GRK5 with ~80%, and finally ΔQ + GRK2 with ~60% (Fig. 4c–f, Supplementary Table 1). The corresponding raw OD values are shown in Supplementary Fig. 2 d-g, demonstrating that these signals fall within a dynamic range of ~0.65–1.0. Thus, these cell lines allowed us to screen inhibitors of interest for their specificity against individual GRKs. In Fig. 5, the GRK inhibitor profiles of GRK2/3 inhibitors (CCG258747, compound 8h, and compound 101) are shown as an example. All three prevent the phosphorylation of T360/S364 in both ΔQ + GRK2 and ΔQ + GRK3 cells. Among them, compound 8h shows the highest potency with IC$_{50}$ values of log −5.7 and log −5.5, respectively (Supplementary Table 3). We did not observe a reduction of the phosphorylation signal by compound 8h in either ΔQ + GRK5 or ΔQ + GRK6 cells. In contrast, CCG258747 reduced ISO-induced phosphorylation to ~50–55% in ΔQ + GRK5 and ΔQ + GRK6 cells and compound 101 to ~25% in ΔQ + GRK5 at the maximal concentration tested (Supplementary Table 3). Conversely, we tested three GRK5/6 inhibitors (CCG273441, compound 18, and compound 19) in these cell lines, which blocked ISO-induced receptor phosphorylation primarily in ΔQ + GRK5 and ΔQ + GRK6 cells (Fig. 6). Again, the covalent

inhibitor CCG273441 showed the highest potency with IC$_{50}$ values of log −7.3. In addition, treatment with CCG273441 resulted in a ~ 30% decrease in phosphorylation signal in ΔQ + GRK3 cells, while compound 19 led to a ~ 40–60% decrease in phosphorylation signal in ΔQ + GRK2 and ΔQ + GRK3 cells at the maximal concentration tested (Supplementary Table 3).

**Application of GRK inhibitors for studies of receptor function**
Our goal was to demonstrate the applicability of GRK inhibitors for functional receptor studies. Phosphorylation of GPCRs affects not only their signaling but also their trafficking. Therefore, we extended the use of GRK inhibitors to other receptors to investigate how the inhibition of specific GRK classes affects their phosphorylation and internalization. To this end, we chose two selective GRK2/3 and GRK5/6 inhibitors validated by 7TM phosphorylation assay experiments (Figs. 5 and 6, Supplementary Table 3), compound 8h and compound 18, respectively. To achieve inhibition of all four GRK isoforms, compound 8h and compound 18 were used in combination. These two inhibitors were selected based on their ability to completely block either GRK2/3 or GRK5/6 activity at maximal concentrations of 30 μM. Receptor phosphorylation of β2, MOP, S1P1, and V2 was analyzed in Western blots using phosphosite-specific antibodies (Fig. 7, left). Signal intensities were quantified and normalized to 100% agonist stimulation. In β2-expressing cells, preincubation with either compound 8h or compound 18 reduced the phosphorylation signal by 17–25%. This effect was more pronounced in MOP expressing cells where each of these inhibitors blocked agonist-induced phosphorylation by 56–65% (Supplementary Table 4). Phosphorylation of S1P1 and V2 was significantly reduced by

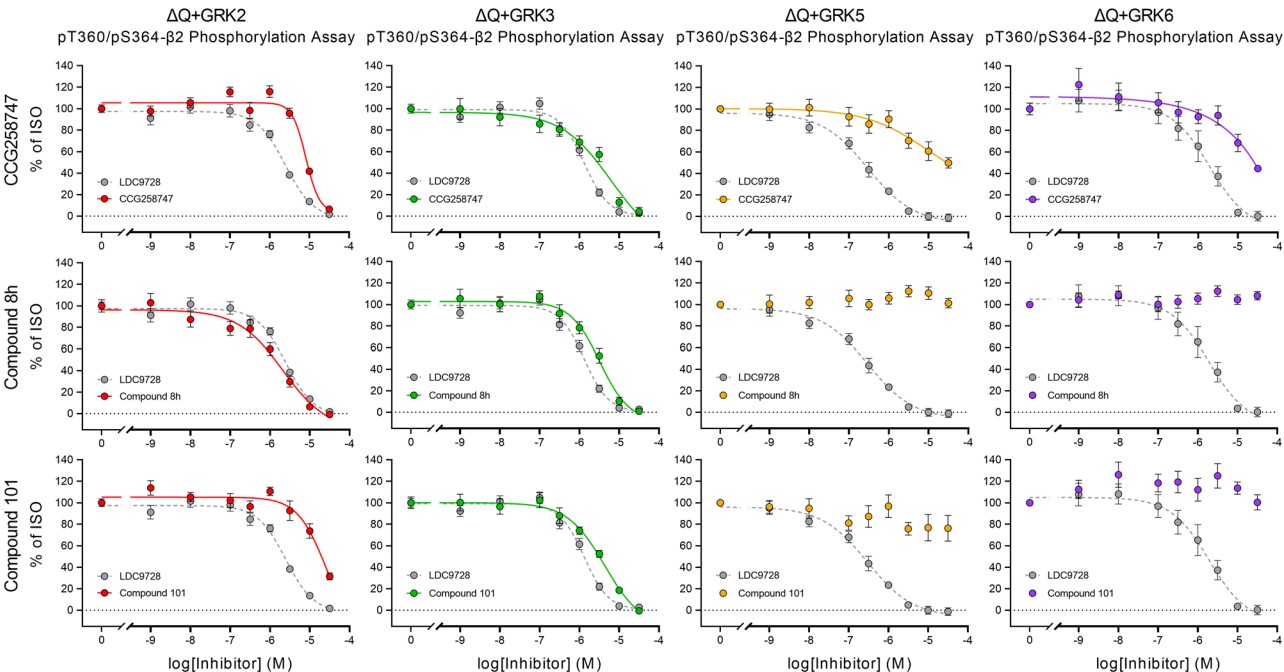

**Fig. 5 | Inhibition of agonist-induced β2 adrenergic receptor (β2) phosphorylation by GRK2/3 inhibitors.** HEK293 ΔQ-GRK cell lines stably expressing the β2 and one GRK isoform (from left to right: ΔQ + GRK2, ΔQ + GRK3, ΔQ + GRK5, ΔQ + GRK6) were preincubated with increasing concentrations of the indicated GRK2/3 inhibitor (CCG258747, Compound 8h, Compound 101) (30 min, 37 °C) before they were treated with 10 µM isoproterenol (ISO) (30 min, 37 °C). Receptor phosphorylation was assessed using the 7TM phosphorylation assay and data were normalized to 10 µM ISO without inhibitor. The pan-inhibitor LDC9728 (dashed in gray line) serves as reference in every graph. Data points represent mean ± SEM of $n = 5$ independent experiments performed in duplicates.

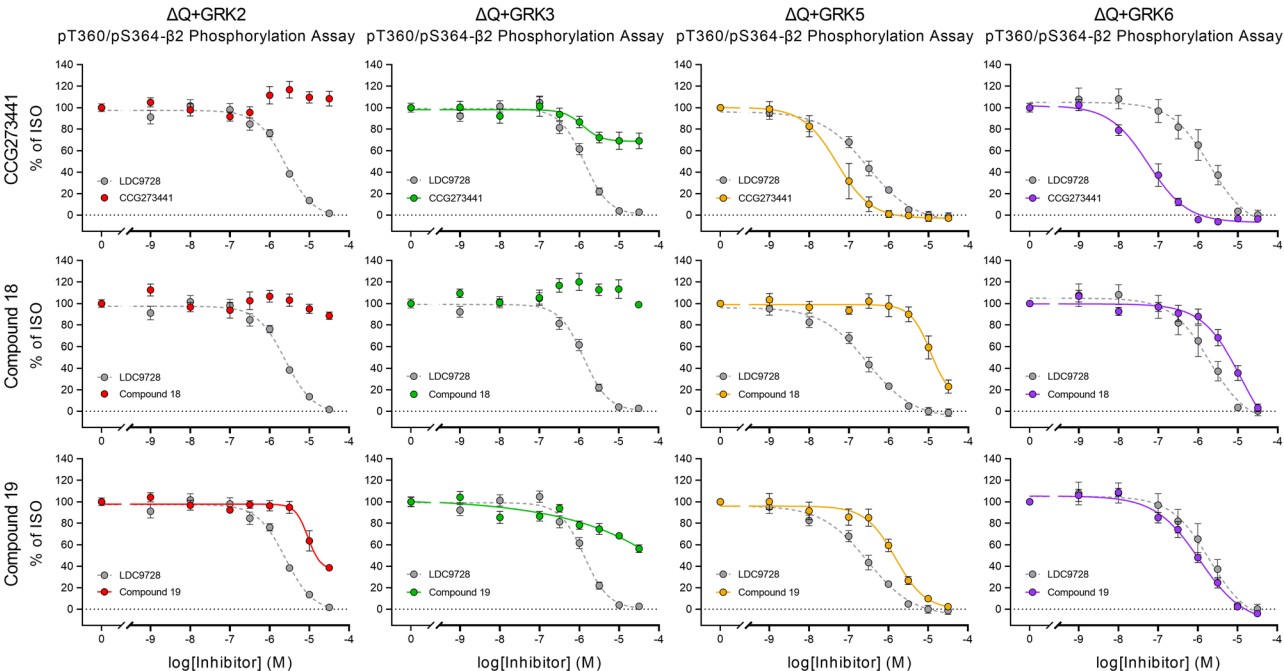

**Fig. 6 | Inhibition of agonist-induced β2 adrenergic receptor (β2) phosphorylation by GRK5/6 inhibitors.** HEK293 ΔQ-GRK cell lines stably expressing the β2 and one GRK isoform (from left to right: ΔQ + GRK2, ΔQ + GRK3, ΔQ + GRK5, ΔQ + GRK6) were preincubated with increasing concentrations of the indicated GRK5/6 inhibitor (CCG273441, Compound 18, Compound 19) (30 min, 37 °C) before they were treated with 10 µM isoproterenol (ISO) (30 min, 37 °C). Receptor phosphorylation was assessed using the 7TM phosphorylation assay and data were normalized to 10 µM ISO without inhibitor. The pan-inhibitor LDC9728 (dashed in gray line) serves as reference in every graph. Data points represent mean ± SEM of $n = 5$ independent experiments performed in duplicates.

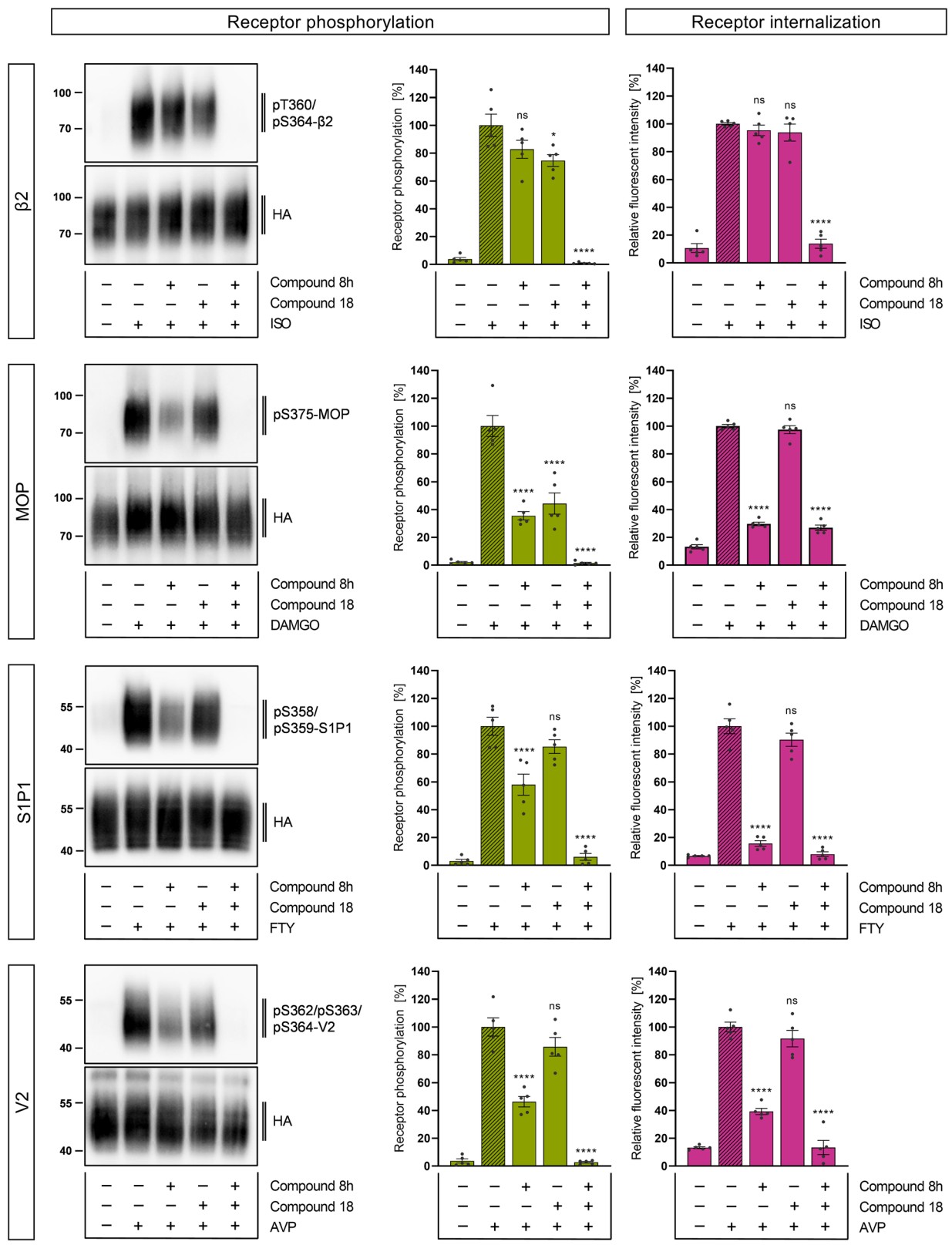

compound 8h, but not by compound 18. However, the combination of both GRK inhibitors completely abolished phosphorylation at all receptors tested (Fig. 7, left). The same conditions were applied to immunofluorescence stainings (Fig. 8) and signal intensities were quantified (Fig. 7, right). Cell morphology appeared to be well preserved under treatment with compound 8h and 18. This is in contrast to what was observed with the even more potent, covalent GRK5/6 inhibitor CCG273441, making it unsuitable for immunocytochemical stainings (Supplementary Fig. 3). Our results illustrate that GPCRs are usually located at the plasma membrane (vehicle control) and internalized upon addition of their respective agonist. In β2-expressing cells, treatment with one of the GRK inhibitors had no effect on agonist-induced internalization. In contrast, receptor internalization of

**Fig. 7 | Influence of Compound 8h and Compound 18 on receptor phosphorylation and internalization.** HEK293 cells stably expressing the β2 adrenergic receptor (β2), the μ-opioid receptor (MOP), the sphingosine-1-phosphate receptor 1 (S1P1) or the vasopressin receptor 2 (V2) were treated with either 10 μM isoproterenol (ISO), 10 μM [D-Ala2,N-MePhe4,Gly-ol]-enkephalin (DAMGO), 10 μM fingolimod (FTY), 1 μM vasopressin (AVP) (30 min, 37 °C) or were left untreated (−). To inhibit certain GRK isoforms, cells were preincubated with 30 μM Compound 8h, 30 μM Compound 18 or a combination thereof (30 min, 37 °C) prior to stimulation with their respective agonist. Receptor phosphorylation of C-terminal residues was imaged by using phosphosite-specific antibodies in Western blot

analysis. Total receptor levels were checked with anti-HA antibody. Western blot images depict one of n = 5 representatives. Signal intensities of all replicates were quantified and normalized to their positive control (agonist stimulation). Receptor internalization was investigated by immunofluorescent stainings. Images from n = 5 replicates were quantified and fluorescence intensities were normalized to their positive control (agonist stimulation). Each data point represents the mean of three cells per replicate. Bar graphs display the mean ± SEM. Statistical differences between conditions were evaluated using one-way ANOVA with Tukey's post-hoc test. ns = not significant; *p < 0.05; **p < 0.01; ***p < 0.001; ****p < 0.0001.

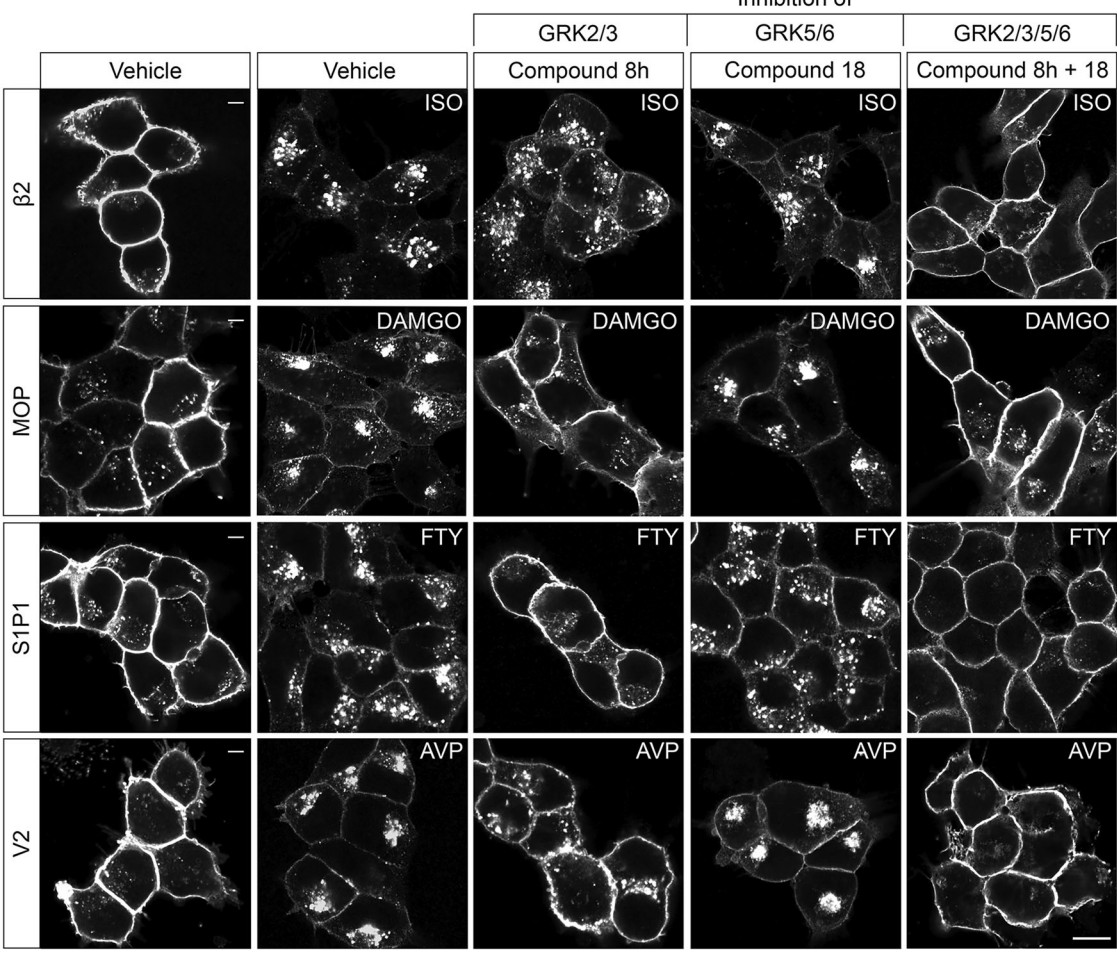

**Fig. 8 | Impact of GRK inhibition on receptor internalization of different GPCRs.** HEK293 cells stably expressing the β2 adrenergic receptor (β2), the μ-opioid receptor (MOP), the sphingosine-1-phosphate receptor 1 (S1P1) or the vasopressin receptor 2 (V2) were treated with either 10 μM isoproterenol (ISO), 10 μM [D-Ala2,N-MePhe4,Gly-ol]-enkephalin (DAMGO), 10 μM fingolimod (FTY), 1 μM vasopressin (AVP) (30 min, 37°C) or were left untreated (−). To inhibit certain GRK

isoforms, cells were preincubated with 30 μM Compound 8h, 30 μM Compound 18 or a combination thereof (30 min, 37 °C) prior to agonist stimulation. Receptor localization was investigated using anti-HA antibody. Immunofluorescent images represent one of n = 5 independent experiments. The scale bar corresponds to 20 μm.

MOP, S1P1, and V2 was significantly reduced by 64-84% after preincubation with compound 8h (Supplementary Table 4). This was not achieved by compound 18 alone. Similar to the Western blot results, a combination of both GRK inhibitors had the greatest effect in all cell lines, achieving levels similar to unstimulated samples (Fig. 7, left). To further illustrate the versatility of the GRK inhibitors, we employed a β-arrestin2 recruitment assay at two additional GPCR systems: the PTH1 and M5, which are differentially regulated by GRKs[41]. Correspondingly, β-arrestin2 recruitment to these receptors was inhibited upon incubation with specific GRK inhibitors, as measured in a BRET assay. While β-arrestin2 recruitment to the GRK2/3-specific receptor (M5) was only diminished by inhibitors targeting GRK2/3, for the GRK2/3/5/6-regulated GPCR (PTH1) it was also affected in presence

of GRK5/6 inhibitors (Supplementary Fig. 4). These results demonstrate the effectiveness of the inhibitors across different receptor types and assay formats.

## Discussion

GRKs are integral to the regulation of GPCR signaling and exert diverse biological functions. Overexpression of certain GRK isoforms has been documented in a variety of pathological conditions, including cancer and cardiovascular diseases. Particularly the GRK isoforms 2 and 5 have attracted attention due to their involvement in β-adrenergic receptor desensitization and regulation of gene expression via histone deacetylase phosphorylation, respectively, which both contribute to the progression of

heart failure[5,42–48]. A dysregulation of GRK6 is associated with various cancers[49–51] and it has been proposed as a potential anti-tumor target in multiple myeloma[17,37,52,53]. This has stimulated interest in the development of GRK inhibitors. Computational modeling and crystallography are key methods for designing small-molecule inhibitors and optimizing kinase-inhibitor interactions. In addition, they allow prediction of inhibitor selectivity and potency, often supported by empirical data from in vitro assays[35–37,54–57]. However, the effects of GRK inhibition cannot be adequately captured by in vitro systems alone, highlighting the value of cellular assays that provide a more physiologically relevant context to assess the impact of GRK modulation on GPCR activity.

This study was designed to develop cell-based GRK inhibitor assays for each individual GRK isoform. To achieve this, we focused on the T360/S364 phosphorylation of the β2. We showed that this site is phosphorylated in a GRK-dependent manner upon a 30-min ISO treatment. Although peak receptor phosphorylation may already occur at earlier time points[58], the incubation times used here represent a balance between biological relevance and experimental feasibility. A large fraction of maximal receptor phosphorylation was retained even when one GRK subfamily was deleted (ΔGRK2/3 or ΔGRK5/6) (Fig. 1, Supplementary Table 1), indicating that T360/S364 can serve as a substrate for all relevant isoforms, including GRK2, 3, 5, and 6 (Fig. 4). Functional redundancy among GRKs has been demonstrated in Drube, et al.[41]. In this study, they introduced a comprehensive panel of GRK knockout cells. Despite the deletion of individual or combined GRKs, residual β-arrestin recruitment persisted across multiple GPCR systems, underscoring the robustness of GRK-mediated regulation even in the face of isoform loss. The β2-expressing ΔGRK2/3 and ΔGRK5/6 cells used in this study are based on the HEK293 cell lines characterized by Drube, et al.[41]. Given that their quantitative analyses revealed no significant changes in the expression levels of the residual GRK isoforms, it is likely that the compensation observed in our experiments reflects functional rather than expressional adaptations.

For our investigation, we then used antibodies against this T360/S364 phosphorylation site in conjunction with genome-edited GRK knockout HEK293 cells, which were engineered to co-express β2 and a single GRK isoform. The resulting panel of four individual GRK assays provides a toolbox of next-generation cellular GRK inhibitor assays that facilitate comprehensive medium- to high-throughput evaluation of the specificity and selectivity of current and novel inhibitors. Previous studies used a variety of surrogate assays like β-arrestin recruitment, internalization, or cell viability. However, such techniques can only provide indirect information and are not suitable for assessing individual GRK isoforms. Thus, the technology reported here represents a substantial advance by allowing the direct measurement of the activity of individual GRKs in a cellular environment.

To demonstrate the utility of our adapted assay, we compared the specificity and potency of a variety of commercially available compounds. Among the GRK2/3 inhibitors, compound 101 has been the most widely used in previous studies. Two compounds, namely CCG258208 and CCG258747, which are structurally derived from paroxetine[34,55], have $IC_{50}$ values in kinase activity assays comparable to that of compound 101[34,55,56]. Under the conditions of our assay, they also showed similar $IC_{50}$ values in ΔGRK5/6 cells. Notably, compound 8h was the most potent GRK2/3 inhibitor tested in this study, yielding an $IC_{50}$ value approximately one order of magnitude lower in ΔGRK5/6 cells compared to the other compounds (Fig. 2, Supplementary Table 2).

Although a variety of GRK2/3 inhibitors are available to date, comparatively few studies have focused on the discovery of novel inhibitors against GRK isoforms 5 or 6. In our study, we tested three inhibitors originally designed by Uehling, et al.[37], namely compound 10a, compound 18, and compound 19. In GRK double-knockout cell lines, we observed improved selectivity and potency of compounds 18 and 19 compared to their parent molecule compound 10a (Fig. 3, Supplementary Table 2), reflecting the authors' progress in optimizing these inhibitors. The most

potent GRK5/6 inhibitor identified was CCG273441, which exhibited by far the lowest $IC_{50}$ value among all compounds tested (Figs. 3 and 6, Supplementary Table 2 and 3). This high potency is likely due to its unique covalent binding mechanism targeting a cysteine residue of GRK5[36]. However, this compound contains a haloacetamide moiety, which is highly reactive and generally toxic in nature. Therefore, current efforts are aimed at developing inhibitors that reversibly bind to GRK5[57]. In our study, we used CCG273441 in immunostaining and observed that incubation with this compound, even at low concentrations, resulted in disrupted cell morphology (Supplementary Fig. 3), probably due to its high reactivity. Therefore, CCG273441 was omitted from subsequent experiments.

In contrast, there were compounds that failed to inhibit phosphorylation in ΔGRK2/3 and ΔGRK5/6 cells at the highest concentrations tested, such as GSK180736A and its analog CCG215022 (Fig. 2, Supplementary Table 2). Previous studies have shown that these GRK inhibitors, as well as other derivatives based on the GSK180736A scaffold, have poor cell membrane permeability and limited membrane retention[34]. Similarly, inhibitors of GRK5/6, namely compound 707 and KR-39038, were unable to block ISO-induced phosphorylation at the maximal concentration tested (Fig. 3, Supplementary Table 2). Although these inhibitors have been used previously in cellular experiments[59–66], we suspect that the incubation times used in our study were insufficient for these compounds to adequately penetrate and accumulate in the HEK293 cell lines.

The only compound that successfully inhibited all GRK isoforms and can therefore be considered a pan-inhibitor was LDC9728[40]. However, its chemical structure has not been published and it is not commercially available. To overcome this limitation, we sought to achieve inhibition of all GRKs by using a combination of two inhibitors. Based on our evaluations, we selected compounds 8h and 18 to confirm their effects on receptor phosphorylation and internalization in Western blot analysis and immunostaining, respectively. Our results show that these inhibitors specifically block phosphorylation in four GPCR-expressing cell lines, which subsequently affects receptor trafficking without compromising cell morphology. These findings are consistent with existing data on the involvement of specific GRKs in receptor regulation[38,41,67–72]. The combination of both inhibitors completely abolished the GRK-mediated GPCR phosphorylation signal, with results comparable to those observed in cell lines lacking expression of all four GRK isoforms (ΔQ-GRK)[40,41,67]. Thus, compounds 8h and 18, as well as combinations thereof, can be recommended as a new gold standard for studies dissecting the functions of GRK2/3 versus GRK5/6 in cellular systems.

It is worth noting that all currently available inhibitors lack complete isoform specificity. In our study, no GRK inhibitor was more than one order of magnitude more potent within the same GRK subfamily. This is not surprising given the high sequence homology between these kinases (GRK2 and GRK3: 84%; GRK5 and GRK6: 70%). Therefore, potential cross-reactivity, whether beneficial or detrimental, must be carefully considered in the design of novel GRK inhibitors.

## Conclusion

In this work, we utilized HEK293 cells co-expressing the β2 and one GRK isoform together with phosphosite-specific antibodies in an immunoassay allowing a thorough assessment of GRK inhibitors. The evaluation of these inhibitors is based on a direct measurement of their ability to prevent receptor phosphorylation. Moreover, our assay enables fast and direct comparison of different commercially available compounds. Our screening revealed that currently available compounds lack complete isoform specificity and can therefore be categorized as GRK2/3, GRK5/6, or pan-GRK inhibitors. Furthermore, we demonstrate the applicability of compound 8h and compound 18 as valuable pharmacological tools to scrutinize the roles of GRK2/3 and GRK5/6. A deeper understanding of GRK function alongside the development of more potent and selective inhibitors holds considerable promise for therapeutic applications targeting GPCR signaling pathways.

## Materials and methods

### Antibodies and reagents

The rabbit polyclonal anti-hemagglutinin (HA) antibody (7TM000HA) as well as the following phosphosite-specific antibodies were provided by 7TM Antibodies (Jena, Germany): pS355/pS356-β2 (7TM0029A), pT360/pS364-β2 (7TM0029B), pS375-MOP (7TM0319C), pS358/pS359-S1P1 (7TM0275C), and pS362/pS363/pS364-V2 (7TM0368C). The anti-rabbit horseradish peroxidase (HRP)-linked antibody (#7074) was obtained from Cell Signaling Technology and the AlexaFluor488-conjugated goat anti-rabbit antibody (#A-11008) from Thermo Fisher Scientific. As agonists served isoproterenol (ISO) (isoproterenol hydrochloride, Merck, I5627, in water) for the β2 adrenergic receptor (β2), [D-Ala$^2$,N-MePhe$^4$,Gly-ol]-enkephalin (DAMGO) (Sigma-Aldrich, E7384, in water) for the μ-opioid receptor (MOP), fingolimod (FTY) (FTY720 (S)-phosphate, MedChemExpress, HY-15382, in DMSO) for the sphingosine-1-phosphate receptor 1 (S1P1), and vasopressin (AVP) ([Arg$^8$]-vasopressin, Tocris, 2935, in water) for the vasopressin receptor 2 (V2). For GRK inhibitor screening, the following substances were purchased: CCG215022 (Seleckchem, S6621), CCG258208 (GRK2-IN-1, MedChemExpress, HY-109562A), CCG258747 (MedChemExpress, HY-139690), CCG273441 (MedChemExpress, HY-47573), compound 8h (GRK2/3 Inhibitor, 7TM Antibodies, 7TMGRK23-IN or GRK2 inhibitor 2, MedChemExpress, HY-156863), compound 10a (GRK6-IN-2, (MedChemExpress, HY-142817), compound 18 (GRK5/6 Inhibitor, 7TM Antibodies, 7TMGRK56-IN or GRK6-IN-1, MedChemExpress, HY-142812), compound 101 (HelloBio, HB2840), compound 707 (GRK5-IN-2, MedChemExpress, HY-136561), GSK180736A (Seleckchem, S8489), and KR-39038 (MedChemExpress, HY-143248) (all in DMSO). Compound 19 (in DMSO) was kindly provided by David Uehling (OICR, Toronto, Canada) and can be purchased at 7TM Antibodies (GRK5/6 Inhibitor 2, 7TMGRK56-IN2). LDC9728 (in DMSO) was provided by Matthias Baumann (LDC, Dortmund, Germany).

### Cell lines and culture

HEK293 cells were originally obtained from DSMZ Germany (ACC 305). GRK knockout cell lines (HEK293 ΔGRK2/3, ΔGRK5/6, and ΔQ-GRK) were established using CRISPR/Cas technology as described by Drube, et al.[41]. Control and knockout cells were transfected with a pcDNA3 plasmid encoding the HA-tagged β2 using polyethyleneimine (PEI; 10 μg/ml, pH 7.2). Cells were selected and maintained in G418-containing medium. Uniform surface expression of the receptor was ensured by FACS sorting using an AF647-conjugated anti-HA antibody, validated by flow cytometry and Western blot analysis. Similarly, stable HA-V2-expressing cells were generated by PEI transfection (10 μg/ml, pH 7.2) and G418 selection[73]. HA-MOP-expressing cells were produced via retroviral transduction as reported by Drube, et al.[41]. To express HA-S1P1 in HEK293 cells, a pcDNA3 plasmid encoding the receptor was transfected using Lipofectamine 2000 according to the manufacturer's protocol. Cells were selected and maintained in G418-containing medium. Single-cell clones were seeded into 96-well plates, and receptor expression was evaluated using the 7TM phosphorylation assay. Expression was further confirmed by Western blot and immunocytochemistry. Stable overexpression of human GRK2-YFP and GRK6-YFP was performed as described in Matthees, et al.[73]. Similarly, stable expression of GRK3-YFP and GRK5-YFP was achieved. In brief, β2-expressing ΔQ-GRK cells were transduced with viral particles made from pMSCV-huGRK3-YFP-IRES-puromycin (ΔQ + GRK3) or pMSCV-huGRK5-YFP-IRES-puromycin (ΔQ + GRK5) vectors. Transduced cells were selected with 0.8 μg/ml puromycin and sorted for consistent YFP fluorescence.

All stable HEK293 lines were cultured in Dulbecco's Modified Eagle Medium (Capricorn Scientific, DMEM-HXA) supplemented with 10% fetal bovine serum (FBS; Capricorn Scientific, FBS-11A) and 1% penicillin/streptomycin (Capricorn Scientific, PS-B). Cells expressing the β2, S1P1, or V2 additionally received 0.5 mg/ml G418 (Capricorn Scientific, G418-B) as selection antibiotic. Cultures were maintained at 37 °C, 5% $CO_2$ and passaged 2–3 times per week. The cell medium was regularly checked for mycoplasma infections using a GoTaq G2 Hot Start Taq Polymerase kit from Promega.

### 7TM phosphorylation assay

The 7TM phosphorylation assay is based on the protocol described by Kaufmann, et al.[40]. HEK293 cells stably expressing the HA-tagged β2 were seeded overnight into poly-L-lysine-coated 96-F-bottom-well cell culture microplates (Greiner Bio-One, 655180). At >95% confluence, cells were stimulated with increasing concentrations of ISO (30 min, 37 °C). For GRK inhibitor experiments, the cell layer was preincubated with a dilution series of CCG215022, CCG258208, CCG258747, CCG273441, compound 8h, compound 10a, compound 18, compound 19, compound 101, compound 707, GSK180736A, KR-39038, or LDC9728 (30 min, 37 °C) prior to treatment with 10 μM ISO (30 min, 37 °C). Cells were then lysed with 150 μl/well detergent buffer (150 mM NaCl; 50 mM Tris-HCl, pH 7.4; 5 mM EDTA; 1% Igepal CA-360; 0.5% deoxycholic acid; 0.1% SDS) containing protease and phosphatase inhibitors (Roche, #04693132001 and #04906845001). A total of 100 μl of cell lysate was transferred into 96-U-bottom-well assay plates (Greiner Bio-One, 650101), dividing it equally into two corresponding wells (50 μl each), which were subsequently required for parallel detection of the phosphorylation signal and the loading control. To enrich the protein of interest, HA-tagged receptors were immunoprecipitated with mouse anti-HA magnetic beads (Thermo Fisher, 88837). Primary antibody dilutions of anti-pT360/pS364-β2 (7TM Antibodies #7TM0029B-PA) or anti-HA (7TM Antibodies #7TM000HA) were added overnight. An HRP-linked anti-rabbit antibody (Cell Signaling Technology) served as secondary antibody and allowed the induction of a colorimetric reaction after adding Super AquaBlue detection solution (Thermo Fisher, 00-4203-58) to each well. This reaction was stopped with 0.625 M oxalic acid, and the supernatant without magnetic beads was transferred to a 96-well F-bottom detection plate (Greiner Bio-One, 655182). Optical density (OD) at 405 nm was determined using the FlexStation3 microplate reader (Molecular Devices) and data were acquired with SoftMax Pro 5.4 software. Calculations were performed in Excel 16.0. Background signals (wells without primary antibody) were subtracted from raw OD values. Afterwards, phosphorylation signals were normalized to their corresponding loading control as well as to the positive and negative control on the same test plate. Concentration-response curves were generated using GraphPad Prism 9.3.1 software and represent mean ± SEM of $n = 5$ independent experiments performed in duplicates.

### Assessment of assay linearity

To evaluate the linearity of the 7TM phosphorylation assay, we performed a dilution series based on mixed cell lysates. HEK293 ΔQ-GRK cells stably expressing the HA-tagged β2 and one GRK isoform or wild-type (WT) HEK293 cells were seeded into poly-L-lysine-coated 96-F-bottom-well cell culture microplates. The next day, cells were stimulated with 10 μM isoproterenol (30 min, 37 °C) and lysed with 150 μl/well detergent buffer containing protease and phosphatase inhibitors. After centrifugation, the supernatants of each cell line were pooled. Lysates of β2-expressing cells and WT cells were aliquoted into 96-U-bottom-well assay plates to produce a dilution series with controlled amounts of β2 per well. The resulting samples, corresponding to 0–100 μl/well lysate from β2-expressing cells, were processed using the standard phosphorylation assay protocol. Receptor phosphorylation was detected using the anti-pT360/pS364-β2 antibody. Raw OD values were corrected for background and bar graphs were plotted from $n = 4$ independent experiments in duplicates. The assay displayed a linear response within ~10–70 μl for all four GRK-overexpressing cell lines (Supplementary Fig. 1), confirming that phosphorylation measurements fall within a quantitative detection window under the conditions used in this study.

### Western blot analysis

HEK293 stably expressing the HA-tagged β2, MOP, S1P1, or V2 were seeded in poly-L-lysine-coated 60-mm dishes (Greiner Bio-One, 628160)

and grown until a confluence of >90% was reached. Cells were exposed to their respective agonist (10 μM ISO, 10 μM DAMGO, 10 μM FTY, or 1 μM AVP; 30 min, 37 °C) or were left untreated. For GRK inhibition experiments, cells were preincubated with 30 μM compound 8h, 30 μM compound 18, or a combination thereof (30 min, 37 °C) prior to agonist stimulation. Afterwards, they were lysed with detergent buffer (150 mM NaCl; 50 mM Tris-HCl, pH 7.4; 5 mM EDTA; 1% Igepal CA-360; 0.5% deoxycholic acid; 0.1% SDS) supplemented with protease and phosphatase inhibitors (Roche). Receptors were enriched with mouse HA epitope-tag antibody agarose beads (Thermo Fisher, 26182) and eluted by incubating in SDS sample buffer (25 min, 50 °C). The supernatant was then loaded onto an 8% polyacrylamide gel for electrophoresis and subsequently immunoblotted onto a PVDF membrane using a semidry electroblotting system (Bio-Rad Trans-Blot® Turbo™ Transfer System). After blocking, membranes were incubated with anti-pT360/pS364-β2 (1:400), anti-pS375-MOP (1:400), anti-pS358/pS359-S1P1 (1:600), or anti-pS362/pS363/pS364-V2 (1:300) to detect receptor phosphorylation or anti-HA (1:750) (7TM Antibodies) to confirm equal protein amounts in every sample. An HRP-linked anti-rabbit secondary antibody (1:5000) (Cell Signaling Technology) allowed visualization of the signals via a chemiluminescence detection system (Thermo Fisher, 34075). Western blot signals were imaged by the Fusion FX7 imaging system (Peqlab). Figures depict one representative of $n = 5$ independent experiments. Uncropped blot images are provided in the Supplementary Information (Supplementary Figs. 5 and 6). Quantification was performed using the software ImageJ. The signal intensity was calculated by subtracting the background signal from the mean intensity of the selected band area. Afterwards, the ratio of the phosphorylation signals to their respective loading control was taken and all data points were normalized to the positive control (agonist stimulation). Calculations were performed in Excel 16.0. Graphs were generated with GraphPad Prism 9 and depict the mean ± SEM of $n = 5$ independent replicates.

### Immunocytochemistry

HEK293 cells stably expressing the HA-tagged β2, MOP, S1P1, or V2 were seeded onto poly-L-lysine-coated coverslips in 24-well plates overnight. When a confluency of 80% was reached, cells were preincubated with anti-HA antibody (1:1000) for 2 h at 4 °C. They were treated with 30 μM compound 8h, 30 μM compound 18, or a combination thereof (30 min, 37 °C) prior to stimulation with their respective agonist (10 μM ISO, 10 μM DAMGO, 10 μM FTY, or 1 μM AVP; 30 min, 37 °C). Cells were then fixed with 4% paraformaldehyde for 30 min at RT, washed with phosphate buffer (22.6 mM $NaH_2PO_4 \cdot H_2O$; 77.4 mM $Na_2HPO_4 \cdot H_2O$; 0.1% Triton X-100; pH 7.4) and permeabilized with 50% and 100% ice-cold methanol for 5 min each. After two additional washing steps, the cell layer was blocked with phosphate buffer containing 10% normal goat serum for 2 h at RT. Afterwards, the cells were incubated with AlexaFluor488-conjugated goat anti-rabbit antibody (1:5000) (Thermofisher Scientific) at 4 °C overnight. The following day, cells were washed several times with phosphate buffer and the coverslips were mounted with Fluoromount-G™ (Invitrogen) including DAPI. Specimens were examined and imaged using a Zeiss LSM900 laser-scanning confocal microscope equipped with the Airyscan 2 technology and ZEN 3.0 software (Zeiss). Fluorescence intensity was determined using ImageJ by determining the integrated density within a defined region of interest (ROI)[74]. This ROI was selected within the cell body, excluding the plasma membrane, to specifically assess the intracellular fluorescence. The plasma membrane was identified based on cell morphology and deliberately avoided when defining the ROI. To correct for background noise, one ROI outside the cell was measured in each image. Three cells were selected from $n = 5$ independent replicates for each condition. The corrected total cell fluorescence (CTCF) was calculated from the integrated density minus background fluorescence and the mean fluorescence intensity was determined per biological replicate. All CTFC values were then normalized to their positive control (agonist stimulation). Calculations were performed in Excel 16.0 and graphs were generated with GraphPad Prism 9. The normalized data are depicted as mean ± SEM.

### Intermolecular bioluminescence resonance energy transfer (BRET) measurements

Recruitment of β-arrestin2 to the muscarinic acetylcholine receptor 5 (M5) and the parathyroid hormone 1 receptor (PTH1) was performed as described in Drube, et al.[41]. Briefly, HEK293 cells were transfected with 0.5 μg M5 (C-terminally tagged with a NanoLuciferase), 1 μg β-arrestin2 (C-terminally fused with a Halo-Tag), and 1 μg empty vector, or 1.4 μg PTH1R (C-terminally fused with a Halo-Tag), 0.35 μg β-arrestin2 (C-terminally tagged with a NanoLuciferase), and 0.75 μg empty vector in 21 cm² dishes following the Effectene transfection reagent manual by Qiagen (#301427). After 24 h, 40,000 cells per well were seeded into a poly-D-lysine-coated white 96-well plate (Brand, 781965) in presence of Halo-ligand (ratio 1:2000; Promega, G980A). Each transfection was seeded in triplicate, and a mock labeling condition (without Halo-ligand) was included. After 24 h, cells were washed twice with measuring buffer (140 mM NaCl; 10 mM HEPES; 5.4 mM KCl; 2 mM $CaCl_2$; 1 mM $MgCl_2$; pH 7.3). Subsequently, cells were incubated with indicated concentrations of GRK inhibitor (Compound 101, CCG273441 or LDC9728) for 30 min, and then the NanoLuciferase substrate furimazine (ratio 1:3,500; Promega, N157B) was added. Measurements were conducted using a Synergy Neo2 plate reader (BioTek) with Gen5 software (version 2.09), employing a custom-made filter (excitation: 541–550 nm, emission: 560–595 nm, fluorescence filter: 620/15 nm). After detecting the baseline for 3 min, indicated concentrations of agonist were added and measurements were continued for 5 min. To correct for labeling efficiency, the mock labeling condition was subtracted from the initial BRET ratio. For Halo correction, the post-ligand stimulation values were divided by the corresponding baseline values. Technical replicates were averaged. For normalization, Δ net BRET change of 1 was defined as 0% and the mean of the agonist-stimulated condition not treated with GRK inhibitor was defined as 100% for each receptor. Calculations were conducted utilizing Excel 16.0 and normalization was performed using GraphPad Prism 9.

### Statistics and reproducibility

In the 7TM phosphorylation assay, each condition was assessed in at least $n = 4$ independent experiments performed in duplicates to ensure reproducibility. Western blot and immunocytochemistry experiments were replicated $n = 5$ times. Quantified data from Western blot and immunocytochemistry images were analyzed statistically using GraphPad Prism 9. Due to the small sample size ($n = 5$), normality was assessed visually using QQ plots. All data points were approximately normally distributed. Statistical differences between the conditions were evaluated using one-way ANOVA followed by Tukey's post-hoc multiple comparisons test.

### Reporting summary

Further information on research design is available in the Nature Portfolio Reporting Summary linked to this article.

### Data availability

The authors declare that the data supporting the findings of this study are available within the paper and its Supplementary Information. Source data are provided in the Supplementary Data sheets. Information regarding plasmids used in this study, as well as any raw data files in other formats are available from the corresponding author upon reasonable request.

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

## Acknowledgements
We thank Ulrike Schiemenz for her excellent technical assistance as well as Monique Brendel for generating stable GPCR-expressing HEK293 cells. Samples of small molecule GRK inhibitors have been kindly provided by Matthias Baumann (LDC, Dortmund, Germany).

## Author contributions
S.S. initiated the project and designed all experiments with N.K.B. and M.K. N.K.B. performed 7TM phosphorylation assays and immunoblot experiments as well as the analysis and presentation of the data. M.K. prepared images and quantification of all immunocytochemical stainings. A.D. contributed 7TM phosphorylation assay data. L.K. and E.S.F.M. designed and performed BRET measurements. F.N. generated and purified phosphosite-specific antibodies used in this study. V.W., J.D., and C.H. engineered and provided all GRK-knockout and GRK-overexpressing cell lines. B.J. and D.U. contributed critical compounds. The manuscript was written and revised by N.K.B. and S.S. with input from the other authors.

## Funding

## Competing interests
S.S. is the founder and scientific advisor of 7TM Antibodies GmbH, Jena, Germany. F.N. is an employee of 7TM Antibodies. All other authors declare

no competing interests. An invention disclosure related to the assay described in this work has been filed at the University Hospital Jena.
