## [Transparent Peer Review file · Communications Biology]

Cell-based and isoform-selective G protein-coupled receptor kinase assays for comprehensive inhibitor evaluation

Corresponding Author: Professor Stefan Schulz

Version 0:

Reviewer comments:

Reviewer #1

(Remarks to the Author)

In this manuscript, the authors used HEK293 cells genetically engineered to co-express the $\beta 2$ adrenergic receptor and one of the four GRK isoforms on a GRK2/3/5/6 knockout background. The authors then tested the inhibition by various GRK inhibitors of isoproterenol-induced T360/S364- $\beta 2$ phosphorylation and suggested that this cell-based assay that should facilitate the screening of GRK inhibitors as potential drugs.

I have the following comments or suggestions:

Throughout the manuscript I could not figure out if there is any genetic modification of the transfected recombinant beta2 receptors? Any modification at T360/S364? Do endogenous beta2 receptors in HEK293 cells play a role?? Have the authors tried to express only GRK without recombinant beta2 receptors?

Although the authors mentioned that the grk inhibition assay is based the 7TM phosphorylation assay established by Kaufmann, Blum, it should be useful to elaborate and briefly recapitulate the rationale in the current manuscript, since the main topic is grk inhibition.

L121-122: change "as reported by Drube, Haider" to "Drube et al."? same with L128: Matthees et al., and elsewhere throughout the manuscript. Please check if this is the right format for ref citing.

L135-337: "Cells co-expressing $\beta 2$ and one GRK isoform, as well as S1P1- and V2-expressing lines, additionally received 1% G418 (Capricorn Scientific, G418-B) as selection antibiotic."

Please check if 1% G418 is correct. Is it 1g/100ml G418? Kind of too much.

Table 1: 3 significant numbers should be sufficient.

Reviewer #2

(Remarks to the Author)

I. Brief summary of the manuscript

The goal of this work was to develop an assay system that allows researchers to carry out high (medium?) throughput screening of cell-permeable isoform-specific GRK inhibitors. This paper builds off the reagents and findings from two recent papers from the Schulz/Hoffman labs. First, in Drube et al 2022, genomic knockouts of the ubiquitously expressed GRK2, GRK3, GRK5 and GRK6 isoforms were prepared in HEK293 cells in many combinations. Important this work are the GRK2/GRK3 and GRK5/GRK6 subfamily-specific double knockouts as well as the quadruple knockout. In Kaufman et al 2024, these labs developed a 96-well plate intact-cell bead-based phosphorylation assay for the mu-opioid, c5a, D1 dopamine, and somatostatin receptors. In the work submitted here, they have used the stably transfected quadruple knockout HEK293 cells stably transfected with beta-2- adrenergic receptor 2 (b2) along with a single GRK isoform and have adapted the 96-well plate assay for the detection of a single phosphosite on the b2 (and perhaps the S1P1 and V2) receptors. Double knockouts assessed the contribution of the two subfamilies of GRKs on the agonist-induced b2 phosphorylation and a battery of commercially available GRK inhibitors were screened. Single isoform transfectants

assessed potency and selectivity (relative to other GRKs) of the inhibitors. This system clearly identified GRK2/3- and GRK5/6-specific inhibitors as well as a general GRK inhibitor as the authors suggest. The GRK2/3- and GRK5/6-specific inhibitors were used to assess intact cell phosphorylation and internalization of b2 and three other GPCRs stably transfected in HEK293 cells. These results demonstrate unambiguous receptor- and GRK-subfamily specific behavior.

II. Overall impression of the work

The regulation of GPCR signaling by ligands, G proteins, GRKS, arrestins and other proteins, all whose expression differs from cell to cell, is an extremely complex process. That the Hoffman/Schulz labs are addressing the role of GRKs in a systematic way is important. This system should allow the identification of more potent and hopefully isoform-specific GRK inhibitors that can cross a cell membrane. The authors demonstrated specificity of the 8h and 18 compounds for targeting GRK2/3- and GRK5/6, respectively, which should allow researchers to quickly assess the roll of GRK subfamilies in the regulation of their favorite receptor. In general I believe that this is very nice work and hopefully that it can provide a standardized path for how any lab might assess the role of GRKs on a receptor in a particular cell line and generate pharmaceuticals to knock down GRK activity in pathological states. In order for this approach to be adaptable to other systems more controls and descriptions might be included. A fuller comparison to the literature is warranted with respect to other intact cell phosphorylation studies, the screens for GRK inhibitors and the results of experiments utilizing GRK inhibitors relative to their results.

III. Specific comments, with recommendations for addressing each comment

1. This work focuses on GRK phosphorylation of the b2 receptor. Vilardaga et al., 2003 first showed that GRKs act seconds after agonist addition in intact cells. Tran et al, 2004, using an intact cell phosphorylation assay, showed that for some phosphosites in the b2, phosphorylation was maximal at 5 minutes. A 30 minute agonist treatment is used for all these studies. I would like to see a time course of phosphorylation at the T360/S364 phosphosite and a rationalization of the 30 minute treatment to study GRK phosphorylation.

2. In both GRK subfamily knockout cell lines, ~70% of the Emax is retained when the other subfamily is deleted (Table 1). Authors might address this phenomenon in the discussion. It would be interesting to know if there is any compensatory GRK expression in subfamily knockout cell lines. Immunoblots in the Drube et al 2022 demonstrate the successful knock out but did not quantify expression of the remaining GRKs. (Most GRK signals looked saturating).

3. To facilitate extrapolating this system to other receptors and specifically to immunoprecipitate HA-tagged receptors of interest, it would be useful to know the level of HA-tagged receptor used in the studies with stably transfected b2 and an GRK isoform. What was the ratio of receptor to GRK? Were the levels set to match any particular cell type?

4. The authors report on a 96-well assay to detect b2 phosphorylation in intact cells. These assays depend on phosphosite antibodies which vary from lot to lot. The authors should describe their approach to assessing linearity in the bead-based assay and demonstrate that the phosphorylation assays are indeed linear. Moreover the range of the results before normalization should be indicated.

5. Despite having developed a bead- and enzyme-based quantitative intact cell phosphorylation assay, the authors utilized immunoblotting to assess receptor phosphorylation in Figure 7. Whys wasn't the bead-based assay used? In my lab, immunoblotting is linear only over a 10-fold range with the readout being insensitive at low levels and saturating with signals that are published. Again, the authors should demonstrate the linearity of the assay if they are presenting quantitative data.

6. For figures 7 & 8, the authors should more fully describe how plasma membrane and internalized receptors were distinguished and quantified with ImageJ.

7. This is very nice work, but as for its novelty, the authors should justify their claim. Tran et al 2004 (Richard Clark lab) used an intact cell immunoblotting assay to measure b2 phosphorylation. Asghar et al 2022 used high-throughput fluorescence immunocytochemical assay to measure delta opioid receptor induced ERK phosphorylation. This assay could potentially be used to screen GRK inhibitors. Many folks have worked with the commercially available GRK inhibitors.

8. The authors should discuss in more detail about the selection of the T360/S364 phosphosite. Did they actually look at other (ex: S355, S356) phosphosites and determine that they were inferior?

9. In Figure 1 especially, but also in some experiments presented in Figures 3 & 4, it looks like more data points are warranted around the predicted EC50 or IC50.

10. The GPCR-arrestin BRET assay needs more introduction in the text – why was it performed? In particular why were the M5 muscarinic and parathyroid hormone used in these experiments rather than b2 and V2 receptors utilized elsewhere in this study?

Version 1:

Reviewer comments:

Reviewer #1

(Remarks to the Author)

I do not have further comments.

Line 27, need to confirm that this is not an error: "compound 8h (GRK2/3 inhibitor) and compound 8h (GRK5/6 inhibitor) are highly"

Reviewer #2

(Remarks to the Author)

Response to authors.

1. Perhaps the authors should acknowledge include the sentence, 'although peak phosphorylation may occur earlier (Tran et al 2004), the chosen incubation times represent a balance between biological relevance and experimental feasibility.'
2. The reader is asked to accept that although the western blotting signals in Drube et al., 2002 Fig 1a look saturating (and thus non-quantifiable), the levels of GRKs in knockout cells lines were in fact rigorously quantified in separate experiment. This reviewer accepts that.
3. To facilitate the application of the bead-based phosphorylation assay to other receptors, the report of ballpark ng HA-tagged receptor/mg protein in lysate would be useful.
4. This comment was very well addressed. The new data confirm that the bead-based phosphorylation assay is linear over a relatively small range and implies that using this assay with another receptor and phosphosite antibody requires performing similar linearity studies.
- 5 -10. Satisfactorily addressed.

Response to the Reviewers

We thank the reviewers for their valuable comments and suggestions. We have addressed each point thoughtfully and have made corresponding revisions to improve our manuscript. Here, we provide a point-by-point response to the reviewer's comments:

Remarks of Reviewer #1		Replies
1.	Throughout the manuscript I could not figure out if there is any genetic modification of the transfected recombinant beta2 receptors? Any modification at T360/S364? Do endogenous beta2 receptors in HEK293 cells play a role?? Have the authors tried to express only GRK without recombinant beta2 receptors?	Modifications of the receptor have been indicated in the manuscript: We used N-terminal HA-tagged $\beta 2$ (L115), which are schematically represented in the Figures 1a and 4a. The T360/S364 site has not been modified. According to Atwood et al., 2011 there are no significant mRNA levels of ADRB2 present in HEK293 cells. The assay protocol includes immunoprecipitation of HA-tagged receptors with mouse anti-HA magnetic beads (L153-155). Therefore, even if there are endogenous $\beta 2$ receptors present, they cannot be detected with the method used.
2.	Although the authors mentioned that the grk inhibition assay is based the 7TM phosphorylation assay established by Kaufmann, Blum, it should be useful to elaborate and briefly recapitulate the rationale in the current manuscript, since the main topic is grk inhibition.	To address this comment, we added the following statement to the introduction: „Therefore, this assay allowed assessment of GRK inhibitor efficacy in living cells, complementing previous in vitro and in silico approaches.” (L79-80)
3.	L121-122: change “as reported by Drube, Haider” to “Drube et al.”? same with L128: Matthees et al., and elsewhere throughout the manuscript. Please check if this is the right format for ref citing.	The citations have been inserted via a citation program (EndNote). The style has been changed to “Nature”.
4.	L135-337: “Cells co-expressing $\beta 2$ and one GRK isoform, as well as S1P1- and V2-expressing lines, additionally received 1% G418 (Capricorn Scientific, G418-B) as selection antibiotic.” Please check if 1% G418 is correct. Is it 1g/100ml G418? Kind of too much.	We agree that our description is too ambiguous. We added 1% of G418 solution to the cell culture medium. This corresponds to 50 mg/100 ml DMEM. We changed the manuscript text for clarification: “Cells expressing the $\beta 2$, S1P1, or V2 additionally received 0.5 mg/ml G418 (Capricorn Scientific, G418-B) as selection antibiotic.” (L134-136)
5.	Table 1: 3 significant numbers should be sufficient.	We agree that reporting more digits than justified by the experimental precision can be misleading. Accordingly, we have rounded numerical values.
Remarks of Reviewer #2		Replies
1.	This work focuses on GRK phosphorylation of the b2 receptor. Vilardaga et al., 2003 first showed that GRKs act seconds after	We can confirm from Western blot experiments that for the T360/S364 site, the maximal $\beta 2$ phosphorylation signal can be detected after 5 min

	agonist addition in intact cells. Tran et al, 2004, using an intact cell phosphorylation assay, showed that for some phosphosites in the b2, phosphorylation was maximal at 5 minutes. A 30 minute agonist treatment is used for all these studies. I would like to see a time course of phosphorylation at the T360/S364 phosphosite and a rationalization of the 30 minute treatment to study GRK phosphorylation.	agonist exposure (data not shown). In our study, we applied the same incubation times (30 min) for agonists to ensure uniform conditions across all GPCR systems. The preincubation time of 30 min for the GRK inhibitors ensured sufficient cellular exposure. Compounds that did not engage with their target within this time frame were considered as not potent enough in a typical research context. Thus, the chosen incubation times represent a balance between biological relevance and experimental feasibility. We note, however, that these incubation times can be adapted depending on the experimental question.
2.	In both GRK subfamily knockout cell lines, ~70% of the Emax is retained when the other subfamily is deleted (Table 1). Authors might address this phenomenon in the discussion. It would be interesting to know if there is any compensatory GRK expression in subfamily knockout cell lines. Immunoblots in the Drube et al 2022 demonstrate the successful knock out but did not quantify expression of the remaining GRKs. (Most GRK signals looked saturating).	We would like to point out that Drube et al. (2022) provided a quantification of the expression levels of the residual GRKs in their Supplementary Information. We acknowledge that immunoblotting is a semi-quantitative method and therefore not ideal for precise quantification. However, these are the only currently available data on the GRK expression levels in these cell lines. Supplementary Figure 1 of their study illustrates that GRK expression levels of residual GRK isoforms were largely unchanged compared to control cells, with only minor, non-significant variations. Based on that, we consider it likely that the compensation observed in our experiments is not caused by expressional adaptations. We added the following section to the discussion: „We showed that this site is phosphorylated in a GRK-dependent manner upon ISO treatment. A large fraction of maximal receptor phosphorylation was retained even when one GRK subfamily was deleted (ΔGRK2/3 or ΔGRK5/6) (Figure 1, Supplementary Table 1), indicating that T360/S364 can serve as a substrate for all relevant isoforms, including GRK2, 3, 5, and 6 (Figure 4). Functional redundancy among GRKs has been demonstrated in Drube, et al.³⁸. In this study, they introduced a comprehensive panel of GRK knockout cells. Despite the deletion of individual or combined GRKs, residual β-arrestin recruitment persisted across multiple GPCR systems, underscoring the robustness of GRK-mediated regulation even in the face of isoform loss. The β2-expressing ΔGRK2/3 and ΔGRK5/6 cells used in this study are based on the HEK293 cell lines characterized by Drube, et al.³⁸. Given that their quantitative analyses revealed no

		significant changes in the expression levels of the residual GRK isoforms, it is likely that the compensation observed in our experiments reflects functional rather than expressional adaptations.” (L399-412)
3.	To facilitate extrapolating this system to other receptors and specifically to immunoprecipitate HA-tagged receptors of interest, it would be useful to know the level of HA-tagged receptor used in the studies with stably transfected b2 and an GRK isoform. What was the ratio of receptor to GRK? Were the levels set to match any particular cell type?	The cell lines used in these experiments were generated and sorted to obtain comparable expression levels of the HA-tagged $\beta 2$ and comparable, moderate expression levels of each GRK isoform across $\Delta Q+GRK$ cell lines. Protein expression was evaluated by immunoblotting, however, this method does not provide an absolute quantification of the amount of molecules present in the cell. We also want to point out that - because of their different subcellular distributions - local concentrations of individual GRKs to the receptor can still differ despite comparable total expression levels. Our aim was not to match any specific cell type in particular, but rather to establish a controlled system that enables direct comparison of GRK-dependent phosphorylation and inhibitor effectiveness. After evaluating GRK inhibitor performance within this standardized context, these substances can subsequently be applied in other models of interest.
4.	The authors report on a 96-well assay to detect b2 phosphorylation in intact cells. These assays depend on phosphosite antibodies which vary from lot to lot. The authors should describe their approach to assessing linearity in the bead-based assay and demonstrate that the phosphorylation assays are indeed linear. Moreover the range of the results before normalization should be indicated.	The antibody concentration is approximately 0.5 $\mu g/\mu l$ (determined by Bradford assay), although we acknowledge that minor variations between lots can occur. In our experiments, however, we did not observe any substantial fluctuations in signal intensity or development time, even when using antibodies from different lots at the same dilution. To further address the reviewer’s concern, we added a supplementary figure demonstrating the linearity of the bead-based immunoassay (L169-184). The assay displayed a linear response between ~10-70 μl for the GRK-overexpressing cell lines (Supplementary Figure 1). In all functional experiments, we used 50 μl of cell lysate per well, thus confirming that our measurements fall within the quantitative detection window (L150-153). Furthermore, we included raw data prior to normalization to illustrate the range of the OD values (Supplementary Figure 2). The difference between unstimulated and stimulated samples varied from ~0.65-1.0 across the GRK-overexpressing cell lines, supporting a sufficient dynamic range that ensures valid interpretation of the obtained data (L326-327).

5.	Despite having developed a bead- and enzyme-based quantitative intact cell phosphorylation assay, the authors utilized immunoblotting to assess receptor phosphorylation in Figure 7. Whys wasn't the bead-based assay used? In my lab, immunoblotting is linear only over a 10-fold range with the readout being insensitive at low levels and saturating with signals that are published. Again, the authors should demonstrate the linearity of the assay if they are presenting quantitative data.	The main goal of the experiments shown in Figure 7 and 8 was to demonstrate additional applications of the GRK inhibitors. Western blotting, a well-established and widely used method for detecting (phosphorylated) proteins, was employed as a complementary approach for the immunostaining data. The blots were intended to illustrate qualitative and semi-quantitative effects. To minimize concerns regarding linearity, blot exposures were selected to avoid saturation and quantification was performed only in non-saturated images. Overall, the bead-based phosphorylation assay still remains our primary quantitative platform.
6.	For figures 7 & 8, the authors should more fully describe how plasma membrane and internalized receptors were distinguished and quantified with ImageJ.	We added information regarding the quantification to the Methods section: “Fluorescence intensity was determined using ImageJ by determining the integrated density within a defined region of interest (ROI). This ROI was selected within the cell body, excluding the plasma membrane, to specifically assess the intracellular fluorescence. The plasma membrane was identified based on cell morphology and deliberately avoided when defining the ROI. To correct for background noise, one ROI outside the cell was measured in each image. Three cells were selected from n=5 independent replicates for each condition.” (L225-231)
7.	This is very nice work, but as for its novelty, the authors should justify their claim. Tran et al 2004 (Richard Clark lab) used an intact cell immunoblotting assay to measure b2 phosphorylation. Asghar et al 2022 used high-throughput fluorescence immunocytochemical assay to measure delta opioid receptor induced ERK phosphorylation. This assay could potentially be used to screen GRK inhibitors. Many folks have worked with the commercially available GRK inhibitors.	We acknowledge that other methods exist that allow the detection of phosphorylated proteins and that commercially available GRK inhibitors have been investigated in various systems. In this study, our aim was to demonstrate how our bead-based immunoassay, in combination with suitable antibodies and cell lines, can be adapted to systemically compare multiple GRK inhibitors under uniform experimental conditions. We believe that this approach remains valuable in its own right, as it enables a robust and reproducible assessment of GRK inhibitor efficacy in a cellular system – an aspect that, to our knowledge, has not been reported in this specific context. Nevertheless, to address the reviewer's concern and to avoid unnecessary controversy, we have removed terms like “novel” and “new” from the manuscript.

8.	The authors should discuss in more detail about the selection of the T360/S364 phosphosite. Did they actually look at other (ex: S355, S356) phosphosites and determine that they were inferior?	In our initial experiments, we compared phosphorylation at the S355/S356 and T360/S364 site. Both sites showed similar regulation patterns in response to agonist stimulation. However, the antibody used for detecting T360/S364 phosphorylation yielded a robust signal at lower antibody concentrations, providing a superior signal-to-noise ratio in our hands. For clarification, we added a brief explanation to the Results section. “In this study, we focused on one GRK-dependent phosphorylation site of the β2. Preliminary analyses of pS355/pS356 revealed similar agonist-dependent phosphorylation profiles, suggesting that analysis of this additional site would not have provided further relevant insights for the aims of this study. Furthermore, the antibody against pT360/S364 yielded a superior signal-to-noise ratio allowing robust assessment of receptor phosphorylation (Supplementary Figure 2b).” (L280-285)
9.	In Figure 1 especially, but also in some experiments presented in Figures 3 & 4, it looks like more data points are warranted around the predicted EC50 or IC50.	We thank the reviewer for this suggestion. In our agonist dose-response experiments, we used log(0.5) concentration steps. As for the assessment of GRK inhibitors we applied a single, standardized concentration series to allow direct comparison under uniform conditions. While it is true that additional data points around the predicted EC₅₀ and IC₅₀ values could provide finer resolution, we do not expect that such additional measurements substantially change the overall conclusions of our work.
10.	The GPCR-arrestin BRET assay needs more introduction in the text – why was it performed? In particular why were the M5 muscarinic and parathyroid hormone used in these experiments rather than b2 and V2 receptors utilized elsewhere in this study?	The β-arrestin2 recruitment assay was included to demonstrate another experimental system in which GRK inhibitors can be applied. We deliberately chose to include GPCR systems other than the β2 and V2 used in the main study illustrating that the inhibitors are effective across different receptor types. The PTH1 and M2 were selected because the working group performing these experiments had extensive experience with these receptors in this assay and expected them to yield robust and reliable signals. We added the following sentences to the Results section: “To further illustrate the versatility of the GRK inhibitors, we employed a β-arrestin2 recruitment assay at two additional GPCR systems: the PTH1 and M5, which are differentially regulated by GRKs. [...] These

results demonstrate the effectiveness of the inhibitors across different receptor types.” (L372-380)

The following Supplementary Figures were added in response to reviewer’s comments:

Supplementary Figure 1: Linearity of the 7TM phosphorylation assay. HEK293 ΔQ-GRK cell lines stably expressing the β2 and one GRK isoform or wildtype (WT) HEK293 cells were stimulated with 10 μM isoproterenol (30 min, 37°C) and lysed with detergent buffer. Different dilutions ranging from 0 μl to 100 μl were generated by mixing lysates from β2-expressing and WT cells. Receptor phosphorylation at the T360/S364 site was assessed using the standard 7TM phosphorylation assay protocol in ΔQ+GRK2 (a), ΔQ+GRK3 (b), ΔQ+GRK5 (c) and ΔQ+GRK6 (d). Raw optical density (OD) values were corrected for background and bar graphs display mean ± SEM from n=4 independent experiments performed in duplicates.

Supplementary Figure 2: Raw optical density (OD) data obtained from HEK293 Control and Δ Q-GRK cell lines overexpressing one GRK isoform. **(a)** Schematic representation of the β 2 adrenergic receptor (β 2) indicating the antibody binding sites targeting two intracellular phosphorylation sites. **(b)** HEK293 cells stably expressing β 2 were stimulated with increasing concentrations of isoproterenol (ISO) (30 min, 37°C). Phosphorylation of the S355/S356 and T360/S364 site were assessed using the 7TM phosphorylation assay. **(c-f)** Graphs show the raw OD values corresponding to the data presented in Figure 4. Concentration-response curves from HEK293 Control and Δ Q-GRK cells are compared with Δ Q+GRK2 **(c)**, Δ Q+GRK3 **(d)**, Δ Q+GRK5 **(e)** and Δ Q+GRK6 **(f)**. **(g)** Bar graph depicts the mean raw OD values from each experiment obtained using the anti-HA antibody, therefore representing the receptor expression in the respective cell lines. All graphs display the mean \pm SEM of raw OD values corrected for background and before normalization of at least n=5 independent experiments performed in duplicates.

Response to the Reviewers

We thank the reviewers again for their comments and suggestions. Here, we provide a point-by-point response to the reviewer's comments:

Remarks of Reviewer #1		Replies
1.	Line 27, need to confirm that this is not an error: "compound 8h (GRK2/3 inhibitor) and compound 8h (GRK5/6 inhibitor)"	Thank you for pointing out this mistake. We changed the abstract to "compound 8h (GRK2/3 inhibitor) and compound 18 (GRK5/6 inhibitor)"
Remarks of Reviewer #2		Replies
1.	Perhaps the authors should acknowledge include the sentence, 'although peak phosphorylation may occur earlier (Tran et al 2004), the chosen incubation times represent a balance between biological relevance and experimental feasibility.'	We included this sentence into the discussion: "To achieve this, we focused on the T360/S364 phosphorylation of the β 2. We showed that this site is phosphorylated in a GRK-dependent manner upon a 30-min ISO treatment. Although peak receptor phosphorylation may already occur at earlier time points (Tran et al., 2004), the incubation times used here represent a balance between biological relevance and experimental feasibility. " (L402-406)
3.	To facilitate the application of the bead-based phosphorylation assay to other receptors, the report of ballpark ng HA-tagged receptor/mg protein in lysate would be useful.	The detection limit of the 7TM phosphorylation assay was determined in a previous study using the μ -opioid receptor (Kaufmann et al.). Therein, we showed a linear increase in signal intensity between 80 pg to 1,200 pg HA-tagged receptor protein per 100 μ l cell lysate. We would expect a similar sensitivity for the β 2 in the current study.